# Adolescents’ Life Satisfaction: The Role of Classroom, Family, Self-Concept and Gender

**DOI:** 10.3390/ijerph17010019

**Published:** 2019-12-18

**Authors:** Amapola Povedano-Diaz, Maria Muñiz-Rivas, Maria Vera-Perea

**Affiliations:** 1Department of Education and Social Psychology, Pablo de Olavide University, 41013 Seville, Spain; mverper1@upo.es; 2Department of Social Anthropology, Seville University, 41013 Seville, Spain; mmrivas@us.es

**Keywords:** family climate, classroom climate, adolescence, self-concept, life satisfaction, gender

## Abstract

This study analyzes the direct relationships between classroom and family context and adolescent students’ life satisfaction (LS) and the indirect relationships between these same variables through adolescents’ academic, family, and social self-concept from a gender perspective. In the theoretical model, we assume that the quality of the parent—child relationship affects adolescents’ LS both directly and indirectly through their self-concept. We assume that the quality of the classroom climate also affects adolescents’ LS through their self-concept. The sample consisted of 2373 adolescents (49.8% girls) aged 12 to 18 years (mean (*M*) = 14.69, standard deviation (*SD*) = 1.82). A structural equation model was tested to analyse the relationship between the variables. Subsequently, multigroup analysis was performed to determine the structural invariance of the model as a function of gender. The chi square and T-student test was 71.66. Results revealed a direct positive relationship between family environment and LS. Family and classroom environment were indirectly related to LS through their relationship with academic, family, and social self-concept. The result of multigroup analysis supports the structural invariance of the model in both sexes; therefore, the expected relationships are the same for boys and girls, making the model more generalizable and applicable. The practical and theoretical implications are discussed.

## 1. Introduction

Nosce te ipsum. Know thyself. This simple sentence inscribed on the walls of the Temple of Apollo at Delphi (Greece) has been a source of inspiration for great philosophers and thinkers of all time in their reflections on happiness. Likewise, the pursuit of happiness and achieving “a good life” are considered to be one of the fundamental missions of psychology [1,2,3]. For the great philosophers, thinkers, and psychologists, self-knowledge is one of the key components in the pursuit of happiness and can help build a healthy self-concept.

From the viewpoint of psychology, happiness studies carried out in recent decades revolve around the construct of subjective well-being (SWB) [4,5]. From the works of Diener et al. [6] in the 1980s, a rich field of research on SWB was developed with a broad consensus about its three fundamental defining characteristics: (1) the experience of the individual’s well-being, his/her perceptions and assessments of well-being in different life areas (e.g., family satisfaction or satisfaction at work); (2) positive emotional responses (e.g., happiness or optimism) and not just the absence of negative ones (e.g., sadness or rage); and (3) an overall appraisal of satisfaction with life (LS) [6,7]. Considered the key indicator of SWB, LS is a cognitive, subjective, and comprehensive assessment of the person’s quality of life [8,9]. High scores on measures of LS are often considered an indicator of happiness [5,10].

The study of adolescents’ happiness has been an area of great interest for the model of positive development and competences in adolescence developed in the 1990s [11,12]. The model of Positive Youth Development has contributed much to the considerable increase of research on LS in children and adolescents in recent years [13,14]. During this distinctive stage, biological, psychological, cognitive, and social changes may affect the evaluation of LS, and self-concept may be considered an indicator of how adolescents cope with these changes in relation to LS [15].

The self-concept during adolescence has been widely studied, particularly in relation to academic achievement [16]. However, studies on the relationship between self-concept and LS during adolescence are relatively recent [17,18]. A possible explanation for the delay in the development of research in this area could be the scientific community’s lack of consensus about the operationalization of the self-concept construct and its conceptual relationships with other variables such as self-esteem and self-efficacy [19]. To avoid confusion, this paper defines the self-concept as the perception of oneself from conscious awareness. In addition, the debate on the dimensionality of the self-concept construct may have influenced the generation of knowledge in this area. We emphasize that multidimensional assessments of the self-concept offer more sensitive, specific, and adjusted measurements [20,21] than non-specific and global measures provided by one-dimensional models [22]. In fact, most of the self-concept research generated since the 1980s uses multidimensional measures [20].

Regarding the psychological and social areas, the self-concept has been more strongly related to psychosocial adjustment in adolescence than to well-being or happiness [22,23].

Psychosocial adjustment is understood by most researchers as the personal and social adaptation of youth [24] or as a psychological and social maturation and well-being during adolescence [25,26]. Research on adjustment outcomes indicates that people’s self-perception predicts LS, subjective well-being, and self-acceptance [17,21]. That is, adolescents with high scores in self-concept perceive their lives more positively [18], whereas a low self-concept is related to negative assessments of one’s life, family, and school [27,28].

Numerous studies also show that family climate has a strong relationship with adolescents’ well-being [29,30,31], behaviour [32,33], social, physical, emotional, and intellectual development, and with the formation of their self-concept [27]. Family climate refers to the psychosocial and institutional family characteristics and its settings [34] and can be conceptualized as the perception and interpretation of the inner world of the family and its relationships [35]. A positive family climate promotes cohesion [30], support, trust, and closeness among family members, favouring empathic and open family communication dynamics [36,37], and offers psychosocial resources to adolescents for the construction of a healthy self-concept [36,38].

Similarly, classroom climate is understood as the social environment based on the degree of academic engagement, the relationship among peers, and the interaction with the teacher in the classroom. Therefore, the combination of subjective perceptions shared by students and teachers about the characteristics of the school setting and of the classroom is important [35,39]. A positive classroom climate is identified when students feel accepted and valued, can express their feelings and opinions freely, and engage in classroom activities [40]. A positive classroom climate promotes students’ emotional, social, and psychological adjustment at school and is related to SL and to the self-concept [40,41,42].

Regarding the involvement of gender in these relationships, recent studies have observed gender differences in some of the variables included in the study. Thus, for example, for adolescent girls, the factors of affectionate relations and adequate communication with their parents have been more closely linked to promoting their psychosocial adjustment than for boys [43,44]. In addition, it should be noted that girls tend to be more integrated socially at school, feel closer to the teaching staff and show better psychosocial adjustment [30,44]. With respect to LS and self-concept in adolescence, in the few studies carried out, the results are not conclusive [13,15]. In this study, we also incorporate gender in the analysis of the relationships among these variables.

Bearing in mind that the works that incorporate self-concept in the investigation of satisfaction with life are still incipient, our research question is: will self-concept have a buffer effect in the relationships between family and classroom climates and satisfaction with life?

From the previous theory, the first objective is to analyse the relationships between family and classroom climate and adolescent’s LS as a function of gender. Therefore, based on the literature review, the following research hypotheses were proposed:

**Hypothesis** **1** **(H1).**
*It was expected that family climate would have a direct and indirect positive relationship with LS through self-concept.*


**Hypothesis** **2** **(H2).**
*It was expected that classroom climate would have an indirect, but not direct, relationship with LS through its direct and positive relationship with social and academic self-concept.*


**Hypothesis** **3** **(H3).**
*It was expected that the relationships proposed would differ significantly depending on gender.*


## 2. Methods

### 2.1. Sample and Procedure

Participants in this study were 2399 adolescents enrolled at nine Compulsory Secondary Education (ESO) schools in western Andalusia (Spain). For the selection of the sample, random group sampling was used in western Andalusia (Spain). The primary units of the sample were the rural and urban areas. The secondary units were private and public schools. Classes were not treated as tertiary units, as all classes were included in our research. The sample is representative of the Andalusian educational community, which numbered 377,574 secondary students in the 2016–2017 academic year. A sample error of ±2.5%, a 95% confidence level, and a population variance of 0.50 were assumed. The size of the sample required was 1531 students. The selection of participants was carried out using multi-stage stratified random sampling [45]. The sample units were rural and public schools in Andalusia. The strata were established by province and school ownership. The final sample consisted of 2373 adolescents of both genders (49.8% female) aged between 11 and 18 years (mean (*M*) = 14.6, standard deviation (*SD*) = 1.78) after excluding 26 students (1.45%) for the following reasons: acquiescence in responses (12); comprehension difficulties (foreign students) (7); voluntary abandonment of the research (3); and failure to obtain parental consent (4). The missing data on the scales were obtained using the method of regression imputation. This method assumes that the rows of the data matrix constitute a random sample of a normal multivariate population.

After obtaining the relevant institutional (school administration) permissions and parents’ written signed consent, trained researchers carried out the survey in the classroom during regular class times in two different sessions of approximately 45 min. It was stressed that participation in the research was voluntary, anonymous, and required prior parental consent. The study met the ethical values required for research on human beings, respecting the basic principles included in the Helsinki Declaration (informed consent and a right to information, protection of personal data, and guarantees of confidentiality, non-discrimination, gratuity, and the option of leaving the study at any time). Data for this research were compiled as part of a broader study on violent behaviour in adolescents in Spain (Reference: PSI2012-33464).

### 2.2. Instruments

*Life satisfaction.* The Spanish version of the 5-item Life Satisfaction Scale [46] was used. The items provide an overall rating of LS in terms of subjective well-being (e.g., “My life is, in most aspects, as I would like it to be”) on a 7-point response scale ranging from 1 (*strongly disagree*) to 7 (*strongly agree*). The confirmatory factor analysis (CFA) performed showed the measurement model had a good fit to the data: the Satorra-Bentler scaled chi-squared statistic test, (SB χ^2^ )= 22.0433, *p* < 0.001, *df* = 4; the confirmaroty fit index (CFI) = 0.99, the root mean square error of approximation (RMSEA) = 0.043, 90% confidence index (CI) (0.02–0.04)). The scale showed acceptable reliability (Cronbach α = 0.76), composite reliability (ρ_c_ = 0.79 and Ω = 0.83), and average variance extracted (AVE = 50%).

*Self-concept*. The “Autoconcepto Forma A” (AFA) (self-concept scale) [47] scale consists of 30 items rated from 1 (*never*) to 5 (*always*), which measure self-concept perception. For the present study, we used the subscales of Social self-concept, Academic self-concept, and Family self-Concept. The Social self-concept (6 items, α = 0.75, ρ_c_ = 0.80, Ω = 0.87, AVE = 50%) combines two aspects, one referring to the ease or difficulty in maintaining or expanding one’s social network (e.g., “I make friends easily”), the other referring to individual qualities that are important for interpersonal relationships (e.g., “I am a happy boy/girl”). The Academic self-concept (6 items, α = 0.88, ρ_c_ = 0.83, Ω = 0.88, AVE = 56%) combines two aspects, one of specific qualities valued in school, (e.g., “I do my homework properly”), other about teachers’ reactions (e.g., “My teachers consider me to be a good student”). The Family self-concept (6 items, α = 0.78, ρ_c_ = 0.80, Ω= 0.85, AVE = 50%) also combines two aspects, one about specific qualities valued in the family, such as involvement, trust, and helping (e.g., “My family would help me in any kind of problem”), the other about parents’ reactions (e.g., “My parents criticize me a lot”). The CFA showed a good fit of the proposed measurement model: SB χ^2^ = 1617.8899, *p* < 0.001, *df* = 345; CFI = 0.91, RMSEA = 0.039, 90% CI (0.037–0.041).

*Family Climate.* The Family Climate Scale [48] is composed of 90 true/false items measuring social and environmental characteristics of families. In this study, the subscale Relationships was used. It consists of 27 items that measure three dimensions: (1) Cohesion or the degree of commitment and family support perceived by the children (e.g., “In my family, we really help and support each other”; 9 items, α = 0.85, ρ_c_ = 0.87, Ω = 0.89, AVE = 50%); (2) Expressivity or the degree to which emotions are expressed within the family (e.g., “In my family, we comment our personal problems”; 9 items, α = 0.80, ρ_c_ = 0.86, Ω = 0.89, AVE = 50%), and (3) Conflict or the degree to which anger and conflict are expressed among family members (e.g., “In my family, we criticize each other frequently”); 9 items, α = 0.86, ρ_c_ = 0.87, Ω = 0.89, AVE = 50%). The CFA showed an acceptable fit of the proposed measurement model: SB χ^2^ = 538.9130, *p* < 0.001, *df* = 4; CFI = 0.92, RMSEA = 0.031, 90% CI (0.027–0.034). The overall reliability of the scale was acceptable (α = 0.85, ρ_c_ = 0.85, Ω = 0.86, AVE = 50%).

*Classroom Climate*. The Classroom Environment Scale [48] was used. This scale is composed of 30 items that assess the social climate and interpersonal relationships within the classroom, with true/false response options. The instrument measures three dimensions: (1) Involvement (e.g., “Students pay attention to what the teacher says”; 10 items, α = 0.73, ρ_c_ = 75, Ω= 0.76, AVE = 50%); (2) Teacher’s support (e.g., “The teacher shows interest in the students”; 10 items, α = 0.74, ρ_c_ = 0.87, Ω = 0.86, AVE = 50%); and (3) Friendship (e.g., “Many classmates become friends in this classroom”; 10 items, α = 0.78, ρ_c_ = 0.79, Ω = 0.79, AVE = 50%). The CFA showed an acceptable fit of the proposed measurement model: SB χ^2^ = 1075.6623, *p* < 0.001, *df* = 4; CFI = 0.91, RMSEA = 0.033, 90% CI (0.030–0.035). The overall reliability of the scale was acceptable (α = 0.80, ρ_c_ = 0.85, Ω = 0.86, AVE = 50%).

### 2.3. Data Analysis

Firstly, Pearson correlations were calculated between all the variables under study, and the analysis of the differences of means according to gender (*t*-test for independent samples). A structural equation model was then tested using the Structural Equations Modeling Software (EQS 6.0) (Multivariate Software, Inc., Temple City, CA, USA) [49] to analyse the relationship between the variables. Robust estimators were used to determine the goodness of fit of the model and the statistical significance of the coefficients, as the coefficient of the normalised estimator showed that there was no multivariate normality [50]. Lastly, a multigroup analysis was carried out to confirm the structural invariance of the model as a function of gender. For this purpose, two models were compared: one with constrictions (which assumed that the relationships between variables are the same for boys and girls) and another without constrictions (which estimates all the coefficients in both groups). An expression by Satorra and Bentler (1994) [51], which allows for scaling of the statistical difference test was used to compare the two nested models.

## 3. Results

Before calculating the structural equation model, zero-order correlation analysis, as well as the means and standard deviations for all the variables, were computed. As Table 1 shows, SL was significantly and positively related to all the dimensions of self-concept, family climate, and classroom environment. Boys scored higher on Cohesion (*r* = 0.50, *p* < 0.01), Involvement (*r* = 0.18, *p* < 0.01), Friendship (*r* = 0.22, *p* < 0.01) and Academic self-concept (*r* = 0.29, *p* < 0.01). Girls scored higher only on Teacher’s support (*r* = 0.20, *p* < 0.01) (Table 1).

Subsequently, a structural equation model was tested. The latent factors included in the model were: (1) Family Climate, with three indicators or observed variables: Cohesion, Expressivity, and Conflict (reversed); (2) Classroom Environment, with three indicators: Involvement, Friendship, and Teacher Support; (3) LS; (4) Familiar self-concept; (5) Social self-concept; (6) Academic self-concept. Table 2 shows the parameter estimates, the number of items loaded on each factor, the standard error, and the associated probability for each observed variable on its corresponding factor. Six factors emerged with a single indicator, presenting a factor load with value 1 and error 0. Bearing in mind that the use of a single measure of global fit of the model is discouraged [50], several indices were calculated: SB χ^2^ = 71.6602, *df* = 21, *p* < 0.001, CFI = 0.99, GFI = 0.99, NNFI = 0.97, AGFI = 0.98, RMR = 0.021, and RMSEA = 0.03 90% CI (0.02–0.04). Values above 0.95 for the CFI, the goodness of fix index (GFI), the non-normed fix index (NNFI), and the adjusted goodness of fit index (AGFI) indices and values below 0.05 for the root mean squared residual (RMR) and the RMSEA are indicative of a good fit. The calculated model fit the data well. This model explained 33.5% of LS (see Table 2).

Figure 1 shows the graphic representation of the final structural model with the standardised coefficients and their associated probabilities. The results showed different relationships of influence between Family Climate, Classroom Climate and LS. Family climate had a direct and positive association with LS (β = 0.34, *p* < 0.001) and also an indirect association, as it had a close positive relationship with Family self-concept (β = 0.66, *p* < 0.001), Social self-concept (β = 0.13, *p* < 0.001) and Academic self-concept (β = 0.16, *p* < 0.001), which, in turn, were directly and positively related to LS (β = 0.15, *p* < 0.001, β = 0.16, *p* < 0.001 and β = 0.09, *p* < 0.001). Family Climate and Classroom Environment were directly and positively related (β = 0.45, *p* < 0.001). Classroom Climate did not have a direct association with LS, but it did present an indirect association. Thus, Classroom Environment had a direct and positive relationship with Social self-concept (β = 0.10, *p* < 0.01) and Academic self-concept (β = 0.20, *p* < 0.001), which, in turn, had a direct and positive relationship with LS (β = 0.16, *p* < 0.001 and β = 0.09, *p* < 0.001).

Lastly, as shown in Table 3, significant differences were found in the multigroup analysis for the non-constrained and constrained models: ∆χ^2^ (14, *N* = 2373) = 25.681, *p* < 0.05. In order to determine which elements of the model generated these differences, the results of the Lagrange multiplier test (ML) provided by the EQS were tested. This test showed that both groups (boys and girls) differed in the path: the association between the latent factor Classroom Climate and its dimension Teacher´s Support was positive in boys (β = 0.47, *p* < 0.001), but it was higher in girls (β = 0.58, *p* < 0.001). After freeing this restriction, the model was statistically equivalent for both genders: ∆χ^2^ (13, *N* = 2373) = 21.36, *p* > 0.05.

## 4. Discussion

The general objective in this research was to analyse the relationships between family and classroom climate and adolescents’ LS, taking gender into account. The results of the analysis of the relationships between these variables indicated that, in general, the climate quality perceived by adolescents, both in the family setting and in the classroom, was related to LS. Although it is well known that both contexts are crucial for boys’ and girls’ psychosocial adjustment, results suggest that the family atmosphere may be more significant for adolescents than the classroom climate. In fact, family climate has a direct and positive relationship with LS, which confirms a part of the first hypothesis proposed. This result is consistent with previous research highlighting the close link between parent–child relations and psychosocial adjustment [32,38,51]. A socio-family environment in which the members feel close ties of affective cohesion, where adolescents can openly express their opinions, emotions, and behaviours to their parents without being judged, and where there are no major sources of conflict favours youths’ self-perception of subjective well-being, happiness, and LS.

In addition, our results suggest that the relationship between the quality of the family climate and adolescents’ LS is also modulated to a significant degree by its influence on their self-concept. Thus, adolescents’ assessment of their family, social, and academic self-concept is intimately related to their assessment of their lives, such that a higher self-concept is associated with high rates of SL. These results confirm the other part of first hypotheses and support the results obtained in other works [15,18,33]. This outcome is very important and helps us understand how self-knowledge and the construction of the self are key elements for adolescents’ happiness. That is, the self-concept during adolescence seems to play an important role in the perception of adolescents’ LS. We hope that the scientific community will return to the study of the self-concept and develop new research that can help us to understand the variables related to young people’s happiness.

The perception of the classroom climate in the classroom also appears to influence adolescent students’ subjective well-being through the social and academic self-concept. Our results indicate that classroom climate has an indirect relationship with LS, through the self-concept, which confirms our second hypotheses. For example, it has been observed that, during adolescence, the quality of social relations in the classroom can influence the psychosocial adjustment of boys and girls, in the sense that the ties of peer friendship are positively associated with a healthy self-concept [19,27,52]. Adolescence is a vital stage in which peer relationships represent a crucial role in the lives of young people. A social network of friends at school provides an essential basis for the emotional development of the individual, and quality friendships enhance the construction of a healthy self-concept [40,53].

Teachers and professors are significant reference figures for young people, as, during this period, they play a key role as mentors in a stage in which young people need to establish a healthy distance from their parents for a positive construction of their identity and their self-concept [44,54]. In the teacher–student relationship, recent studies suggest that, when students feel accepted, valued, respected and listened to by the teacher, the general classroom climate improves, as well as the students’ emotional well-being [19,40]. In terms of gender differences, the degree of student-perceived teacher aid and support is significant both for boys and girls, although girls tend to establish more positive relationships. This result is consistent with previous research indicating that girls tend to be socially more integrated at school [55], closer to teachers and with a better psychosocial adjustment in this context [30,56], thereby confirming the third hypotheses. Finally, it is important to note that although the effect sizes in the multivariate analyses of variance (MANOVAs) and in univariate analyses of variance (ANOVAs) are small, these results seem to maintain their importance in the field.

However, in spite of its positive contributions, this study has some limitations that should be taken into account in the interpretation of the results. Among these limitations, the cross-sectional nature of the study did not establish causal relationships between the studied variables, so longitudinal studies should test these relationships more in depth. In addition, we consider it interesting to address the bi-directional nature of the relationships between the variables included in this research. For example, as we have seen, a good family climate contributes to perceived subjective well-being and, at the same time, adolescents’ high LS can become a facilitator to which parents react by strengthening the positive pattern of family interaction [17,29].

## 5. Conclusions

We believe that our study provides interesting information about the direct relationship of self-concept with adolescents’ adjusted development and subjective well-being. However, it is also important to highlight the role of self-concept as mediator or buffer in the relations between the family setting, especially the school setting, and adolescents’ LS. During this exploratory juvenile stage, the role of loving and supportive parents, and also their surveillance and supervision, are also very important for the development of children’s well-being and happiness [57]. The construction of a self-concept that is consistent with the role that they will play in the adult stage involves boys’ and girls’ exploring the social world, expanding it, and making it more complex and richer. They also need significant adults who, through open, sincere, and fluid communication and through affective social support, help them to adequately manage the conflicts inherent to the development of new relational networks outside the family.

In fact, in the classroom context, the relationship with the teachers and other significant adults has a direct impact on young people’s well-being through the academic self-concept. This result is very revealing and, in our opinion, could have interesting practical implications. For example, strategies to improve LS from the school could contemplate intervention aspects related to the social and academic self-concept, especially in adolescent girls. In addition, our results suggest that successful interventions depend on the relations between the family and LS. Therefore, it would be very appropriate to establish cooperative links between the family and the school, as they are the main immediate social contexts of influence in adolescents’ well-being. Finally, the results of this study raise some interesting questions to address in future investigations that integrate self-concept and the socialization of gender to the variables analyzed when attempting to explain adolescents’ subjective well-being and happiness.

## Figures and Tables

**Figure 1 ijerph-17-00019-f001:**
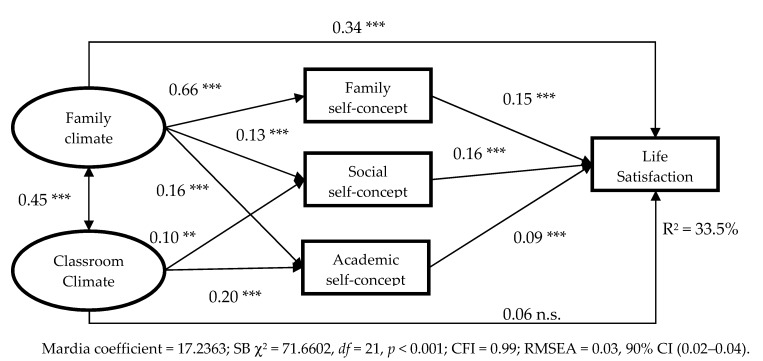
Standardized solution of the model with the correlation coefficients and statistical significance. ** *p* < 0.01. *** *p* < 0.001. SB χ^2^—the Satorra-Bentler scaled chi-squared statistic test; CFI—the confirmaroty fit index; RMSEA—the root mean square error of approximation; CI—the confidence index.

**Table 1 ijerph-17-00019-t001:** Correlation Matrix between the variables of the model and Student *t*-tests as a function of gender (boys on the left of the diagonal).

Variables	*M* (*SD*)	1	2	3	4	5	6	7	8	9	10
1. Cohesion	1.80 (0.21)	-	0.39 **	0.50 **	0.19 **	0.18 **	0.25 **	0.54 **	0.11 **	0.23 **	0.37 **
2. Expressiveness	1.58 (0.20)	0.44 **	-	0.17 **	0.15 **	0.10 **	0.19 **	0.35 **	0.14 **	0.14 **	0.27 **
3. Conflict	1.69 (0.19)	0.54 **	0.19 **	-	0.14 **	0.15 **	0.19 **	0.42 **	0.06	0.15 **	0.32 **
4. Involvement	1.46 (0.21)	0.19 **	0.15 **	0.12 **	-	0.28 **	0.33 **	0.15 **	0.04	0.14 **	0.14 **
5. Friendship	1.72 (0.17)	0.22 **	0.16 **	0.19 **	0.35 **	-	0.26 **	0.09 **	0.12 **	0.10 **	0.14 **
6. Teacher’s supp.	1.60 (0.22)	0.23 **	0.13 **	0.19 **	0.35 **	0.32 **	-	0.20 **	0.02	0.20 **	0.20 **
7. Fam. self-conc.	82.43 (16.68)	0.56 **	0.35 **	0.40 **	0.12 **	0.21 **	0.16 **	-	0.28 **	0.41 **	0.44 **
8. Soc. self-conc.	74.24 (15.10)	0.17 **	0.12 **	0.11 **	0.12 **	0.21 **	0.02	0.27 **	-	0.14 **	0.29 **
9. Acad. self-conc.	61.40 (21.23)	0.20 **	0.17 **	0.14 **	0.17 **	0.14 **	0.14 **	0.41 **	0.18 **	-	0.27 **
10. Life Satisf.	3.31 (0.75)	0.43 **	0.27 **	0.32 **	0.18 **	0.22 **	0.15 **	0.50 **	0.29 **	0.29 **	-
*M* Boys/Girls (Student’s *t*)		1.81/1.80 (1.3)	1.57/1.59 (−1.9) *	18/1.69 (−0.6)	1.46/1.45 (1.5)	1.71/1.72 (−1.6)	1.59/1.61 (−2.6) **	81.63/83.23 (−2.3) *	74.47/74.01 (0.7)	58.85/63.98 (−5.9) ***	3.33/3.28 (1.4)

Note: The correlation is significant at the 0.01 level (two-tailed). Student’s *t*: * *p* < 0.05 (bilateral). ** *p* < 0.01 (bilateral). *** *p* < 0.001 (bilateral). *M*—mean; SD—standard deviaiton.

**Table 2 ijerph-17-00019-t002:** Estimates of parameters, number of items, standard errors and associated probability.

Variables	Number of Items	Factor Loadings
(1) Family climate		
Cohesion	9	1 ^a^
Expressiveness	9	0.59 ** (0.03)
No conflict	9	0.68 ** (0.03)
(2) Classroom climate		
Involvement	10	1 ^a^
Affiliation	10	0.77 ** (0.05)
Teacher’s help	10	1.08 ** (0.07)
(3) Family self-concept	6	1 ^a^
(4) Social self-concept	6	1 ^a^
(5) Academic self-concept	6	1 ^a^
(6) Life Satisfaction	5	1 ^a^

Note: ^a^ set to 1 during the estimate. Significance: ** *p* < 0.01.

**Table 3 ijerph-17-00019-t003:** Multigroup analysis.

Model	Description	SB χ^2^	*df*	Difference SB χ^2^	Difference *df*	*p*
Model 1	Model with restrictions	124.42	56	-	-	-
Model 2	Unconstrained model	98.74	42	25.68	14	<0.05
Model a	1 freed restriction	120.10	55	21.36	13	0.07

Note: SB χ^2^—the Satorra-Bentler scaled chi-squared statistic test; *df*—the degree of freedom; Difference SB χ^2^—the difference between both models in the Satorra-Bentler test; Difference *df*—the difference between both models in degree of freedom. Classroom climate path→Affiliation is positive in males (β = 0.47, *p* < 0.001), but higher in females (β = 0.58, *p* < 0.001).

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
