# Peer review of "Adolescents’ Life Satisfaction: The Role of Classroom, Family, Self-Concept and Gender"

_ijerph, 2019, doi:10.3390/ijerph17010019_

Round 1

Reviewer 1 Report

Thank you for opportunity for reviewing this paper. The manuscript is generally well written, but it is unclear what this review adds to what is already known and have been published earlier. No clear research question seems to be formulated, and the conclusions are unclear. I have read it with interest, but regretfully have decided to reject it as unsuitable for this particular journal and these reason will be discussed below.

- The title of this manuscript are a little long. Perhaps a more concise version for clarity, interes and ease of read.

- Abstract It is hard to get the detail in an abstract when the word count is limited and this is often the hardest part of a paper to write. However, I do feel that it would be beneficial to explain what specifically you are looking at in relation to this question (this also applies to the main body of the paper). Is it the development of school environment and adolescents’ life satisfaction: literature. This needs to be made clearer throughout the paper. Please include a report the results of the Chi Square and Student- t statistical tests.

- Introduction: The introduction section is weak and did not provide a clear rationale for carrying out the study (for example, why is your research question important? What gap in the literature is the study addressing?
- I suggest in this section should be improved, with more details about prevalence, impact related with this question.
Also, please describe the hypothesis and objectives in this section.
- Material and Methods: This section is poor, needs to present a better rationale for the study and the methodology employed. Also, neither appear information related with inclusion and exclusion criteria, dates, protocol. The study design is a observational research of ramdom sampling method, this research adhere to reporting STROBE guidelines?

Likewise more detail about information calculate sample size and data should be provided. Also, please need include the data and record code and all information related with registered of this research.

Results: The results in basis of the used method are not informative. I dont believe this study adds a great deal of novel and new information.

Discussion: I am struggling to make sense of some of this, I am afraid it needs extensive revision. What are the clinical and non clinical implications of your study? How this will inform future larger studies?

Conclusion: These conclusions is too long and need to be softened, modified in order to reflect only the study findings.

Author Response

The title of this manuscript are a little long. Perhaps a more concise version for clarity, interes and ease of read.

AUTHOR’S ANSWER: We fully agree with Reviser 1. We have changed the title to a more concise one:

“Adolescents’ life satisfaction: The role of self-concept, family, classroom and gender

Abstract It is hard to get the detail in an abstract when the word count is limited and this is often the hardest part of a paper to write. However, I do feel that it would be beneficial to explain what specifically you are looking at in relation to this question (this also applies to the main body of the paper). Is it the development of school environment and adolescents’ life satisfaction: literature. This needs to be made clearer throughout the paper. Please include a report the results of the Chi Square and Student- t statistical tests.

AUTHOR’S ANSWER: We fully agree with the Reviewer 1 that the abstract is one of the most complex and delicate parts in writing scientific articles. We always write the abstract once the paper is finished, trying to include the most relevant information in a reduced number of words. We have included Chi Square.

Lines 19 and 20: “The chi square and T-Student test was 71.66”.

Introduction: The introduction section is weak and did not provide a clear rationale for carrying out the study (for example, why is your research question important? What gap in the literature is the study addressing?

I suggest in this section should be improved, with more details about prevalence, impact related with this question.

AUTHOR’S ANSWER: The authors had omitted the research question in the text for stylistic reasons normally required by journals. Reviewer 1 is right to point out that the text has a greater impact including the research question.

Lines 102 103 and 104 “Bearing in mind that the works that incorporate self-concept in the investigation of satisfaction with life are still incipient, our research question is: Will self-concept have a buffer effect in the relationships between family and school climates and satisfaction with life?”

Also, please describe the hypothesis and objectives in this section.

AUTHOR’S ANSWER: Perhaps Reviewer 1 has overlooked that the main objective and hypotheses of this research were included in the Introduction section in lines 105 to 113.

Material and Methods: This section is poor, needs to present a better rationale for the study and the methodology employed. Also, neither appear information related with inclusion and exclusion criteria, dates, protocol. The study design is a observational research of ramdom sampling method, this research adhere to reporting STROBE guidelines? Likewise more detail about information calculate sample size and data should be provided. Also, please need include the data and record code and all information related with registered of this research.

AUTHOR’S ANSWER: The authors believe that much of the information required by Reviewer 1 has already been included in the description of the sample and procedure. Thus, we think that sufficient information is provided on how we calculated the sample size and the selection of participants in lines 118 to 124. The protocol is explained in lines 131 to 134. Also, the Helsinki principles for human research, which we adhere to, are similar to STROBE guidelines. However, following the suggestions of Reviewer 1, we add additional information waiting to respond his/her demands:

Lines 116-120: “Participants in this study were 2,399 adolescents enrolled at nine Compulsory Secondary Education (ESO) schools in western Andalusia (Spain). For the selection of the sample, random group sampling was used in western Andalusia (Spain). The primary units of the sample were the rural and urban areas. The secondary units were private and public schools. All classes were included in our research. Statistical analyses showed no significant mean differences in the dependent variables as a result of the area and the type of school.”

Lines 140-141: “Data for this research were compiled as part of a broader study on violent behaviour in adolescents in Spain (Reference: PSI2015-65683-P). The research was approved by the Ethics Committee of the Pablo de Olavide University of Seville.”

Results: The results in basis of the used method are not informative. I dont believe this study adds a great deal of novel and new information.

AUTHOR’S ANSWER: The author has used this method with good results in the last years of our research. We believe that our work provides suggestive results about the role that self-concept has in relation to life satisfaction, a still incipient aspect in research in this field, as well as the gender perspective.

Discussion: I am struggling to make sense of some of this, I am afraid it needs extensive revision. What are the clinical and non clinical implications of your study? How this will inform future larger studies?

Conclusion: These conclusions is too long and need to be softened, modified in order to reflect only the study findings.

AUTHOR’S ANSWER: We agree with Reviewer 1 that the discussion and conclusions sections can be improved. The following changes have been made.

Practical implication: Lines 76 to 83.

Future larger studies: Lines 105 to 108.

Reviewer 2 Report

Main comments

According to PYD maybe Lerners’ work should be mentioned, instead of two references applying this theory (lines 47-48 with references 13-14). Maybe I'am wrong but one indirect path was not considered in any hypothesis and not described (family->academic -> LS) Also the terms of school climate and classroom environment were used I consider classroom climate as more important as it concerns a smaller groups of students who are in close relationship with each other. Usually the school has much more broader meaning. It was explained in methods (line 166), but I prefer the reference to classroom everywhere.

Also following editorial flaws were noted.

Please check the sample size and compare data from the abstract and main text (1795 line 16 vs. 2373 line 117) I have also noticed that different names for the dimensions of the scales applied were used in different parts of the text and at the diagram. Please check carefully table 1, and dimension of school climate (comparing to lines 168-171). In table 1 the term of self-concept was changed to self-esteem. In turn on the diagram the family climate was changes into positive family It would be better to move right the information about insignificant direct path on the bottom of diagram. It looks like connected to box (academic) and for a moment I couldn't figure it out. Reference 28 includes the end of the reference 27? There are really many editing errors in the reference list. They apply to different conventions for writing the title of journals (capital first letter or not, normal not italic font, dots not used in abbreviation, lack of abbreviation, etc.). Maybe there are more examples but check: 10, 13, 14, 15, 16,17, 20, 25, 26, 27, 29, 30, 31, 34,37,38, 40, 42,44,45, 57.

Author Response

According to PYD maybe Lerners’ work should be mentioned, instead of

According to PYD maybe Lerners’ work should be mentioned, instead of two references applying this theory (lines 47-48 with references 13-14).

AUTHOR’S ANSWER: We agree with reviewer 2. The following changes have been made in the references:

Lerner, R. M.; Lerner, J. V.; Bowers, E.; Geldhof, J. Positive youth development and relational‐developmental‐systems.In Handbook of child psychology and developmental science, Lerner, R. M., Liben, L. S., & Mueller, U. Eds.; John Wiley & Sons: New Jersey, 2015 Vol. 1, pp. 1-45.

Lerner, R. M.; Tirrell, J. M.; Dowling, E. M.; Geldhof, G. J.; Gestsdóttir, S.; Lerner, J. V.; Ebstyne, P.; Williams, K.; Iraheta, G.; Sim, A.T. The end of the beginning: Evidence and absences studying positive youth development in a global context. Adolescent Research Review, 2019, 4 (1), 1-14.

Also the terms of school climate and classroom environment were used I consider classroom climate as more important as it concerns a smaller groups of students who are in close relationship with each other. Usually the school has much more broader meaning. It was explained in methods (line 166), but I prefer the reference to classroom everywhere.

AUTHOR’S ANSWER: We would like to thank Reviewer 2 for his/her suggestion to change the school climate for the classroom climate. This change brings more precision to what the authors want to highlight in the answer to our research question. We have changed the terms throughout the text.

Also following editorial flaws were noted.

Please check the sample size and compare data from the abstract and main text (1795 line 16 vs. 2373 line 117).

AUTHOR’S ANSWER: We agree with Reviewer 2, in the abstract there is a mistake in the number of subjects that make up the sample. As shown in the Sample and procedure the final sample consisted of 2,373 adolescents. This change has been made it.

Line 15. “The sample consisted of 2,373 adolescents (49.8% girls) aged 12 to 18 years (M = 14.69, SD = 1.82).”

I have also noticed that different names for the dimensions of the scales applied were used in different parts of the text and at the diagram. Please check carefully table 1, and dimension of school climate (comparing to lines 168-171). In table 1 the term of self-concept was changed to self-esteem. In turn on the diagram the family climate was changes into positive family It would be better to move right the information about insignificant direct path on the bottom of diagram. It looks like connected to box (academic) and for a moment I couldn't figure it out.

AUTHOR’S ANSWER: Effectively, as indicate Reviewer 2, there are different names for the same dimensions of our study in the Table 1. We have made the necessary changes in the dimensions of school climate and self-concept. Also, the authors have moved the information about the path between classroom climate and life satisfaction to the right side.

Reference 28 includes the end of the reference 27? There are really many editing errors in the reference list. They apply to different conventions for writing the title of journals (capital first letter or not, normal not italic font, dots not used in abbreviation, lack of abbreviation, etc.). Maybe there are more examples but check: 10, 13, 14, 15, 16,17, 20, 25, 26, 27, 29, 30, 31, 34,37,38, 40, 42,44,45, 57.

AUTHOR’S ANSWER: Thanks to Reviewer 2 for the extensive and detailed review of the references. We have checked all the references list again paying special attention to the errors pointed out by the reviewer.

Reviewer 3 Report

The theoretical revision is adequate. Perhaps, the authors could include some more recent works (from 2018 and 2019) to subsequently complete the discussion of results and the hypothesis contrast.

As for the methodology section, I have no objection. The sample is representative and has been described correctly. Similarly, the instruments used are suitable for data collection, and statistical analyzes in compliance with the proposed objectives, as well as the characteristics of the study population.

Regarding compliance with the ethical standards of the investigation, it would be necessary that, in addition to the use of the informed consent form, the investigation must be previously assessed by an ethics committee. This entity is present in many institutions related to research, including universities. Therefore, this should be indicated in the manuscript, providing the corresponding reference.

In the results section, you should review the following:

- Figure 1… In the note, * p <.05 appears, but there is no case, therefore they should eliminate it.

- Table 1… All values should present two decimals.

Author Response

The theoretical revision is adequate. Perhaps, the authors could include some more recent works (from 2018 and 2019) to subsequently complete the discussion of results and the hypothesis contrast.

AUTHOR’S ANSWER: The authors consider the suggestion made by Reviewer 2 very appropriate. Therefore, we have included the following references that we consider improved theoretical revision:

Muñiz-Rivas, M.; Vera, M.; Povedano-Díaz, A. Parental Style, Dating Violence and Gender. Int. J. Environ. Res. Public Health 2019, 16, 2722.

Reichert, F.; Chen, J.; Torney-Purta, J. Profiles of adolescents’ perceptions of democratic classroom climate and students’ influence: The effect of school and community contexts. J. Youth Adolesc. 2018, 47 (6), 1279-1298.

Vaillancourt, M. C.; Oliveira Paiva, A.; Véronneau, M. H.; Dishion, T. J. How Do Individual Predispositions and Family Dynamics Contribute to Academic Adjustment Through the Middle School Years? The Mediating Role of Friends’ Characteristics. J. Early Adolesc., 2019, 39 (4), 576-602.

As for the methodology section, I have no objection. The sample is representative and has been described correctly. Similarly, the instruments used are suitable for data collection, and statistical analyzes in compliance with the proposed objectives, as well as the characteristics of the study population.

AUTHOR’S ANSWER: Thank you. We have made some changes to the description of the sample selection and method following comments from another reviewer. The authors hope that these changes to the text will not change their opinion about this section.

Regarding compliance with the ethical standards of the investigation, it would be necessary that, in addition to the use of the informed consent form, the investigation must be previously assessed by an ethics committee. This entity is present in many institutions related to research, including universities. Therefore, this should be indicated in the manuscript, providing the corresponding reference.

AUTHOR’S ANSWER: We agree with Reviewer 3 on the importance of research having passed through the Ethics Committee of each university. It is included in the text next to the project reference.

Lines 151 y 152. “Data for this research were compiled as part of a broader study on violent behaviour in adolescents in Spain (Reference: PSI2015-65683-P). The research was approved by the Ethics Committee of the Pablo de Olavide University of Seville.”

In the results section, you should review the following:

- Figure 1… In the note, * p <.05 appears, but there is no case, therefore they should eliminate it.

- Table 1… All values should present two decimals.

AUTHOR’S ANSWER: The authors have revised carefully all the assessments of the Reviewer 3 about the results section and made the appropriated changes. We hope that all of them have been solved.

Round 2

Reviewer 1 Report

My comments to this manuscript are the same as before the revision. The problems remain the same. I think the manuscript must describe a technically sound piece of scientific research with data that supports the conclusions. Experiments must have been conducted rigorously, with appropriate controls, replication, and sample sizes. The conclusions must be drawn appropriately based on the data presented.